# Calculating Protein Content of Expressed Breast Milk to Optimize Protein Supplementation in Very Low Birth Weight Infants with Minimal Effort—A Secondary Analysis

**DOI:** 10.3390/nu12051231

**Published:** 2020-04-27

**Authors:** Michaela Minarski, Christoph Maas, Corinna Engel, Christine Heinrich, Katrin Böckmann, Wolfgang Bernhard, Christian F Poets, Axel R Franz

**Affiliations:** 1Department of Neonatology, University Children’s Hospital, Tübingen University Hospital, 72076 Tübingen, Baden-Wuerttemberg, Germany; ch.maas@arcor.de (C.M.); christine_heinrich@gmx.de (C.H.); katrin.boeckmann@med.uni-tuebingen.de (K.B.); wolfgang.bernhard@med.uni-tuebingen.de (W.B.); Christian-F.Poets@med.uni-tuebingen.de (C.F.P.); Axel.Franz@med.uni-tuebingen.de (A.R.F.); 2Center for Pediatric Clinical Studies, University Children’s Hospital, Tübingen University Hospital, 72076 Tübingen, Baden-Wuerttemberg, Germany; Corinna.Engel@med.uni-tuebingen.de

**Keywords:** infant, premature, very low birth weight infant, enteral feeding, nutrition, breast milk, protein supply, individual fortification

## Abstract

Breast milk does not meet the nutritional needs of preterm infants, necessitating fortification. Breast milk is particularly variable in protein content, hence standardized (fixed dosage) supplementation results in inadequate supply. This was a secondary analysis of 589 breast milk protein content measurements of 51 mothers determined by mid-infrared spectroscopy during a clinical trial of higher versus lower protein supplementation in very low birth weight infants. Mothers (and breast milk samples) were divided into a test (41 mothers) and a validation cohort (10 mothers). In the test cohort, the decrease in protein content by day of lactation was modeled resulting in the breast milk-equation (BME)). In the validation cohort, five supplementation strategies to optimize protein supply were compared: standardized supplementation (adding 1.0 g (S1) or 1.42 g protein/100 mL (S2)) was compared with ‘adapted’ supplementation, considering variation in protein content (protein content according to Gidrewicz and Fenton (A1), to BME (A2) and to BME with adjustments at days 12 and 26 (A3)). S1 and S2 achieved 5% and 24% of adequate protein supply, while the corresponding values for A1–A3 were 89%, 96% and 95%. Adapted protein supplementation based on calculated breast milk protein content is easy, non-invasive, inexpensive and improves protein supply compared to standardized supplementation.

## 1. Introduction

Breast milk feeding is the recommended feeding for all newborn infants including those born preterm [1]. Breast milk protects from complications of prematurity (e.g., necrotizing enterocolitis, retinopathy of prematurity, intraventricular hemorrhage or severe infections [2,3,4]). However, breast milk itself does not meet the high nutritional needs of growing preterm infants. Since postnatal growth retardation is associated with a worse neurodevelopmental outcome [5,6,7], supplementation with fortifiers is common practice [8]. Fortification results in higher protein intake and contributes to better short-term growth [9,10]. Breast milk protein content of mothers who delivered preterm is known to decrease from 2.2 g/100 mL (0.3–4.1 g/100 mL) in week 1 to 1.0 g/100 mL (0.6–2.2 g/100 mL) in week 10/11 after birth and shows inter-individual differences [11]. Standardized fixed dosage supplementation, therefore, leads to under- or oversupply with protein depending on the underlying assumption for the average protein content [12] relative to the recommended protein supply (4.0–4.5 g/kg/d weight <1000 g; 3.5–4.0 g/kg/d weight 1000–1800 g [13]). Individual supplementation (adjustable or targeted) results in better protein supply and promotes growth in very preterm infants [14,15,16,17] but also involves additional burden to the children (blood sampling), workload and costs.

The aim of this secondary analysis of the breast milk samples analyzed in a previous randomized controlled trial (RCT) was to establish a feasible supplementation strategy for everyday practice in neonatal units. This adapted supplementation strategy compensates for the variations in breast milk protein content during lactation with minimum extra effort. We present a practical strategy for optimizing protein intake without the need for blood sampling or breast milk measurements, and compared this with standardized protein supplementation based on the breast milk protein content of 598 breast milk samples from 51 mothers obtained during the underlying clinical trial.

## 2. Materials and Methods 

### 2.1. Study Design (Underlying Clinical Study)

The underlying randomized controlled trial published in 2017 [18] was aimed at evaluating the effect of different levels of protein supply on short-term growth. The study protocol was approved by the Institutional Review Board and the trial registered with clinicaltrials.gov (NCT1773902). Written informed parental consent was obtained. The study was conducted in accordance with the Declaration of Helsinki. From October 2012 to October 2014, 60 predominantly breast milk-fed very low birth weight infants were included (randomized in a three-arm study model with standardized low protein intake, standardized high protein intake and individually supplemented breast milk based on the breast milk protein content actually measured) and no significant difference in growth velocity from birth to end of intervention between different levels of protein supply (primary outcome) was shown [18]. Actual protein content of breast milk was measured twice per week by mid-infrared spectroscopy.

This not pre-planned secondary analysis was based on the entire data set of breast milk protein measurements obtained in the main study. The available sample size reflects the needs of the main study; the intervention of the main study did not impact on this analysis.

### 2.2. Study Design (This Secondary Analysis)

#### 2.2.1. Participants

Fifty-two mothers who gave birth to preterm infants with gestational age < 32 weeks and weight < 1500 g were included in the underlying study. One mother stopped expressing breast milk shortly after study entry. All protein content measurements in the expressed breast milk of the 51 remaining mothers were included in this analysis. 

The study population was divided into a test cohort (41 mothers; 457 measurements) to create a regression equation for protein content of preterm breast milk by day of lactation (‘breast milk-equation’) and a validation cohort (10 mothers; 141 measurements) to compare five different supplementation strategies (fixed dosage and adapted supplementation). We assessed the proportion of days with adequate protein intake (defined as 2.7–3.3 g protein/100 mL breast milk).

#### 2.2.2. Measurement of Protein Content 

Breast milk protein content was measured twice weekly (usually Mondays and Thursdays—not on specific days of lactation) using a human milk analyzer (Miris, Uppsala, Sweden; mid-infrared spectroscopy). Calibration was carried out daily as recommended by the manufacturer using a check solution. An aliquot of 5 mL breast milk was used to perform three repeated analyses. Protein content was recorded as the mean of these three measurements. The milk samples used for measurement were each taken from a larger, well-mixed aliquot of milk and were heated to 40 °C before measurement was carried out.

#### 2.2.3. The Breast Milk-Equation

Non-parametric smoothing of the individual data of protein content of the breast milk samples in the test cohort by day of lactation was performed to identify the type of regression equation suitable to model the decrease in protein content resulting in: the breast milk-equation (BME).

#### 2.2.4. Individual Adjustment of the Breast Milk-Equation and Formation of the Above-Mentioned Validation Cohort

We presumed that supplementation would be even more on target if individual adjustments of the breast milk-equation took the actual (measured) protein content of the breast milk into account, thereby compensating for inter-individual differences in breast milk protein content. We assumed that the decline in protein during lactation would be constant; therefore, adjustments for inter-individual differences would require a shift on the y-axis.

Establishment of lactation following preterm delivery requires several days and protein content shows the highest variation during the first 28 days of lactation. Hence, we intended to adjust the breast milk-equation twice during the period from days 7 to 28 after delivery, at least 10 days apart. 

It was arbitrarily decided to adjust the breast milk-equation on days 12 and 26 after delivery since these were the days with the largest number of breast milk samples (16 mothers each). 

To allow comparability of all five supplementation strategies, it was necessary that all mothers within the validation cohort had breast milk measurements on days 12 and 26. Therefore, the first 6 mothers in chronological order after study onset, who had breast milk measurements on days 12 and 26, were assigned to the test cohort. The following 10 mothers, who had breast milk measurements on days 12 and 26, formed the validation cohort.

#### 2.2.5. Target Protein Supply

Adequate protein supply was defined as 2.7–3.3 g protein/100 mL breast milk, resulting in a daily supply of 4.1–5.0 g/kg/d if infants received 150 mL/kg/d according to the ESPGHAN (European Society for Gastroenterology Hepatology and Nutrition) recommendations [13].

#### 2.2.6. Standardized versus Adapted Protein Supplementation Strategies to be Compared

In order to optimize protein supply in very low birth weight preterm infants five different supplementation strategies were compared in the validation cohort (10 mothers; 141 measurements). The primary outcome of this secondary analysis was the rate of breast milk samples reaching target protein supply after supplementation for each of the following 5 supplementation strategies:(S1) Standardized supplementation adding 1 g protein/100 mL(S2) Standardized supplementation adding 1.42 g protein/100 mL(A1) Supplementation based on estimated protein content according to Gidrewicz and Fenton [11] (Table 1)(A2) Supplementation based on protein content calculated by the ‘breast milk-equation’ (Table 1)(A3) Supplementation based on protein content calculated by the breast milk-equation with individual adjustment of the BME according to actual protein content on days 12 and 26.

#### 2.2.7. Statistical Analyses

Statistical analyses were performed using SAS9.4. Non-parametric smoothing of the individual data of protein content of the breast milk in the test cohort by day of lactation was performed to identify the type of regression equation suitable for modeling the decrease in protein content. The regression equation best fitting the data was determined via model fit statistics, resulting in the BME.

To identify differences in the rates of samples that would be below, within and above the target protein range with the different supplementation strategies the generalized linear model approach with weighted least square estimation of parameters was used.

## 3. Results

### 3.1. Demographics

A total of 598 expressed breast milk samples from 51 mothers of 59 preterm infants were included. An overview of the demographic data of the two study groups is shown in Table 2.

### 3.2. The Breast Milk-Equation

#### 3.2.1. Calculation of the Breast Milk-Equation

The BME was established based on protein content measurements in 457 breast milk samples from 41 mothers. Following non-parametric smoothing of the protein content by day after delivery (Figure 1), smoothed data were best described as y = a/x + b. The breast milk-equation best fitting the data was: Protein content [g/100 mL] = 6.755/day after delivery + 0.852.

#### 3.2.2. Individual Adjustments of the Breast Milk-Equation

The breast milk-equation was adjusted twice using the actual protein content on days 12 and 26 after delivery. Adjustment of the BME to individual breast milk protein content on days 12 and 26 was performed by a shift of the breast milk protein content graph on the y-axis, i.e., by adjustment of ‘b’ in the equation y = a/x + b (Table 3).

The performed adjustment and the resulting change in the BME of the mothers assigned to the validation cohort is aditionally shown in (Appendix A).

### 3.3. Comparison of the Supplementation Strategies in the Validation Cohort (10 Patients, 141 Breast Milk Samples) 

**S1:** In 134 (95%) samples, protein supply after supplementation would be lower than 2.7 g/100 mL, in no sample protein supply would be higher than 3.3 g/100 mL and in 7 (4.96%) samples protein supply would meet the target.

**S2:** Protein supply after supplementation in 104 (74%) samples would result in undersupply and in 3 (2.14%) samples in oversupply, while in 34 (24%) samples the protein supply would meet the target. 

**A1:** On 13 (9.29%) days supplementation would lead to undersupply, on 3 (2.14%) days to oversupply and on 124 (88.57%) days supplementation would meet the target. 

**A2:** Supplementation according to the BME in one (0.71%) sample the protein supply would be under 2.7 g/100 mL, in 5 (3.55%) samples over 3.3 g/100 mL and in 135 (95.75%) samples the protein supply would meet the target 

**A3:** Following individual adjustments of the BME on days 12 and 26, protein supply would be too low in 5 (4%) samples; in 2 (1.4%) samples protein supply would be too high and in 134 (95%) samples protein supply would meet the target.

The results of the 5 supplementation strategies are summarized in Table 4.

Adapted protein supplementation resulted in larger proportions of samples on target than standardized supplementation (Table 5). Within the different strategies of adapted fortification, A2 led to a better result than A1, while no difference was found between strategies A2 and A3.

The resulting protein supply during lactation of each mother in the validation cohort after supplementation according to all 5 strategies is also illustrated in (Appendix A).

## 4. Discussion

Standardized supplementation with a fixed dosage of fortifier leads to undersupply of protein caused by decreasing protein content of breast milk during lactation [12]. Standardized supplementation improves short-term growth (if compared to no supplementation) [19,20] and good postnatal growth rates are associated with better neurocognitive outcome in very low birth weight infants [5,21], but fortified milk feeding still resulted in extra-uterine growth restriction [22]. Despite the advantages of an increased protein intake, standardized supplementation with high doses of protein carries the presumably low risk of an oversupply of protein and thus potential side effects such as metabolic acidosis or an increased rate of necrotizing enterocolitis due to hyperosmolar feeding. Although a protein intake of up to 4.6 g/kg/d was not associated with additional side effects [23], a more precise supplementation strategy with reduced risk of under- or oversupply seems necessary.

This analysis shows that consistent with previous findings, standardized supplementation by adding 1 g protein or even 1.42 g protein per 100 mL breast milk would still lead to undersupply in 95% and 74%, respectively, of breast milk-fed-infants. Hence new fortification strategies are needed.

Individualized supplementation (previously entitled ‘adjustable’ [14] or ‘targeted’ [16] fortification) seems an advantageous strategy to prevent protein malnutrition in preterm infants: adjustable fortification adds extra protein to the breast milk controlled by repeated measurements of blood urea nitrogen (BUN). This method, compared to standardized supplementation, leads to improved protein intake and better postnatal growth [14,15]. In addition, ‘adjustable’ fortification takes the individual protein requirements of every infant into account, which can vary due to comorbidities, different enteral resorption and varying metabolic needs. Nevertheless, measuring blood urea nitrogen twice a week leads to extra needle pricks and blood loss. Therefore, non-invasive strategies that result in an improved protein supply would be preferable. 

In ‘targeted’ fortification, frequent measurements of breast milk macronutrients are performed to add individually calculated protein amounts to breast milk, leading to improved protein intake [16,17], but results are inconsistent with respect to growth performance [17,18]. Whereas this is non-invasive, performing frequent milk analyses requires additional time, effort and expensive equipment and may not be feasible in every neonatal unit.

It is, therefore, necessary to establish a method for actual protein content prediction that is both non-invasive and easily feasible. In a systematic review, Gidrewicz and Fenton (2014) included 41 studies to identify breast milk’s protein content of mothers who delivered on term or preterm. Using the data from this meta-analysis and adding protein according to current ESPGHAN recommendations resulted in adequate protein supply in almost 90% of breast milk samples. Even better results were found when using BME derived from the test cohort. Supplementation according to the BME led to a protein supply within the target area (defined as 2.7–3.3 g protein per 100 mL milk) in 96% of breast milk samples of the validation cohort. 

We attempted to further improve the use of the BME by taking the actual protein content of individual mothers at two time points into account (days 12 and 26 after delivery) to compensate for inter-individual variation in protein content. These adjustments, however, did not further improve protein supply. 

This secondary analysis suggests that using the estimated protein content of breast milk according to Gidrewicz and Fenton [11] or according to our BME (which we propose to refer to as ‘adapted’ supplementation) will result in in a substantial increase in days with protein supply in the target area compared to standardized supplementation with 1 or 1.42 g of protein per 100 mL breast milk without burden to the infants or costs. Breast milk samples studied herein were donated by unselected mothers of very preterm infants in a tertiary care neonatal department in a high-income European country, hence the derived BME and the results may apply to this population, and should not be extrapolated to other populations with different food patterns and a higher proportion of malnourished mothers. Generalization of the findings of this secondary analysis is limited due to lacking information about mothers’ nutritional status, comorbidities and dietary habits, which may influence protein content in the breast milk.

In addition, protein content within the target range using the BME has to be confirmed in an independent cohort with analysis of protein content at predefined days of lactation. Furthermore, it would be desirable to study the effect of applying the BME in comparison to other supplementation strategies (e.g., adjustable supplementation) on short-term growth in adequately powered non-inferiority studies. However, compared to adjustable fortification, this strategy does not take the individual needs of preterm infants (due to individual differences in absorption and metabolism) into account. Therefore, determination of blood urea nitrogen (adjustable fortification) or estimation of urinary urea concentrations [24] would be helpful for detecting undersupply resulting from impaired protein absorption or increased protein needs, if preterm infants show growth failure despite adapted protein supplementation.

## 5. Conclusions

‘Adapted’ protein supplementation of breast milk for preterm infant nutrition by using the ‘breast milk-equation’ is an easy, non-invasive and inexpensive way of improving protein supply in preterm infants that resulted in protein supply being on target in >95% of breast milk samples.

## Figures and Tables

**Figure 1 nutrients-12-01231-f001:**
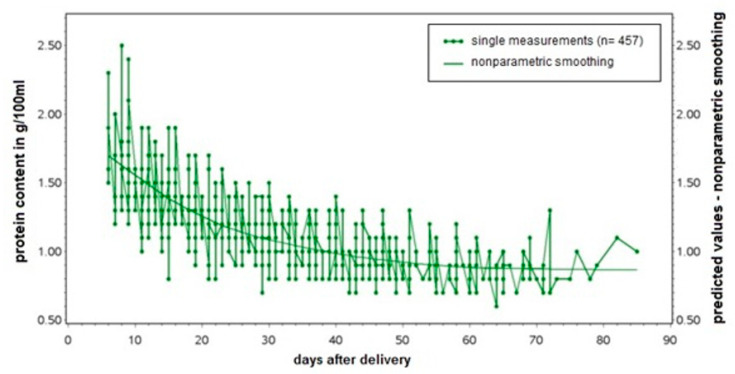
The breast milk-equation (BME): Protein content [in g/100 mL] = 6.755/day after delivery + 0.852.

**Table 1 nutrients-12-01231-t001:** Protein content of breast milk from mothers who delivered preterm in a meta-analysis performed by Gidrewicz and Fenton [11] and resulting supplementation to meet the targeted area of protein supply (2.7–3.3 g protein/100 mL breast milk) compared to protein content of breast milk, calculated by use of the breast milk-equation (g/100 mL breast milk) and resulting supplementation.

Supplementation Acc. to Gidrewicz and Fenton 2014			Supplementation Acc. to Breast Milk-Equation		
Day After Delivery	Estimated Protein Content (g/100 mL)	Resulting Protein Supple-Mentation (g/100 mL)	Day After Delivery	Calculated Protein Content (g/100 mL)	Resulting Protein Supplementation (g/100 mL)
4–7	1.7	1.3	6	2.0	1.0
8–14	1.5	1.5	11	1.5	1.5
15–28	1.4	1.6	21	1.2	1.8
29–42	1.1	1.9	35	1.0	2.0
43–63	1.1	1.9	53	1.0	2.0
64–84	1.0	2.0	74	0.9	2.1

**Table 2 nutrients-12-01231-t002:** Demographic data of patients in the underlying study shown as median (p25–p75) or number in the test and validation cohort.

	Test Cohort	Validation Cohort
	Median (p25–p75); n/n; n	
No. of mothers	41	10
No. of breast milk samples analyzed	457	141
No. of infants	49	10
Gestational age at delivery (weeks)	29.9 (28.6–31.1)	29.6 (28.5–31.1)
Infants’ birth weight (kg)	1.21 (1.09–1.39)	1.07 (0.92–1.28)
Infants’ sex male/female	23/27	4/6
Duration of hospital stay of the infants (days)	40 (30–56)	61 (50–72)
First milk measurement (days after delivery)	8 (7–9)	8.5 (8–11)
No. of milk measurements per mother	11 (8–15)	13 (12–18)

**Table 3 nutrients-12-01231-t003:** Adjustments of the breast milk-equation (BME) in the validation cohort (10 mothers; Pat_adj_1–10) based on actual measured protein content. Protein content calculated for day 12 (1.42 g/100 mL) and day 26 (1.11 g/100 mL) are listed along with the actual measured content and resulting modification of b (‘b after adjustment’) in the equation y = a/x + b.

Validation Mother	b in Original BME	Calculated Protein d12 (g/100 mL)	Measured Protein d12 (g/100 mL)	b After Adjustment d12	Calculated Protein d26 (g/100 mL)	Measured Protein d26 (g/100 mL)	b After Adjustment d26
Pat_adj_ 1	0.852	1.42	1.57	1.007	1.11	1.00	0.740
Pat_adj_ 2	0.852	1.42	1.43	0.867	1.11	1.03	0.770
Pat_adj_ 3	0.852	1.42	1.47	0.907	1.11	1.00	0.740
Pat_adj_ 4	0.852	1.42	1.70	1.137	1.11	1.13	0.870
Pat_adj_ 5	0.852	1.42	1.23	0.667	1.11	1.03	0.770
Pat_adj_ 6	0.852	1.42	1.30	0.737	1.11	1.00	0.740
Pat_adj_ 7	0.852	1.42	1.90	1.337	1.11	1.17	0.900
Pat_adj_ 8	0.852	1.42	1.30	0.737	1.11	1.03	0.770
Pat_adj_ 9	0.852	1.42	1.70	1.137	1.11	1.03	0.770
Pat_adj_ 10	0.852	1.42	1.60	1.037	1.11	1.27	1.010

**Table 4 nutrients-12-01231-t004:** Five supplementation strategies are listed, comparing how often (days (percent)) supplementation resulted in an undersupply (<2.7 g protein/100 mL breast milk), a supply within the target area (2.7–3.3 g/100 mL) or in an oversupply (>3.3 g protein /100 mL breast milk) in the 10 validation mothers.

Supplementation Strategy	<2.7 g/100 mL Protein Supply n (%)	Target of Protein Supply (2.7–3.3 g/100 mL) n (%)	>3.3 g/100 mL Protein Supply n (%)
S1	134 (95)	7 (5)	-
S2	104 (74)	34 (24)	3 (2)
A1	13 (9)	124 (89)	3 (2)
A2	1 (1)	135 (96)	5 (4)
A3	5 (4)	134 (95)	2 (1)

**Table 5 nutrients-12-01231-t005:** Comparison of different protein supplementation strategies in the validation cohort.

Comparison of Meeting the Targeted Area of Protein Supply	% of the Days with Supply in Target Area of Protein Supply	*p*-Value
A2 vs. S1	96% vs. 5%	<0.0001
A2 vs. S2	96% vs. 24%	<0.0001
A2 vs. A1	96% vs. 89%	0.01
A2 vs. A3	96% vs. 95%	0.74

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
