# Peer review of "Calculating Protein Content of Expressed Breast Milk to Optimize Protein Supplementation in Very Low Birth Weight Infants with Minimal Effort—A Secondary Analysis"

_nutrients, 2020, doi:10.3390/nu12051231_

Round 1
Reviewer 1 Report
Dear authors,
This manuscript discusses a novel manner in which to evaluate the protein intake of neonates utilizing a novel equation that the researchers derived. While interesting to the readers and scientific community, there is significant need to improve the scientific rigor of the manuscript in its presentation. My comments are the following:
1) Please ensure the English grammar, syntax, and punctuation are all correct. For instance, line 37 has breast milk capitalized when it should not be. Additionally, there is no period on line 47 to conclude the sentence. These issues arise throughout the manuscript and must be amended.
2) The introduction needs expansion of why protein supplementation is needed in preterm neonates. Notably, the latest Cochrane review discussing the importance of protein supplementation is lacking entirely from this paper and could be incorporated into both the introduction and discussion sections.
3) Additionally, there is no mention of the potential benefits and harms with protein supplementation. Specifically, while it is associated with potentially improve neurodevelopmental outcomes (see Bonnie Stevens et al DOI: https://doi.org/10.1542/peds.2008-0211), there may be potential for increased risk of necrotizing enterocolitis due to increased osmolality of feeds when protein supplementation is added. (Cochrane Review showed a relative risk that was above 1 indicating potential for harm, but was not statistically significant): doi: 10.1002/14651858.CD000433.pub2. These risks and benefits of supplementation are imperative in a manuscript discussing supplementation, and would enhance the discussion section of this manuscript.
4) Lines 46-47 do not appear to belong with the previous paragraph. I would recommend improving this thesis sentence that sets up the entirety of this manuscript to make it more understable to the average reader.
5) The materials and methods section must include a description of what the a priori objectives, especially given that this a secondary analysis. Additionally, what was the outcome the initial power calculation was based on?
6) A citation is needed after "was shown" on line 60.
7) It is unclear initially when reading this methods section how the study was implemented. This needs to be clarified earlier in this section. Specifically, under study design, there must be a description of the study design. Here, that would include the use of an initial test cohort and then a validation cohort subsequently. This is not discussed in the study design but must be included here.
8) Throughout the methods section, there is no description about how the cohorts were devised. Were they paired (in that the same patients underwent different supplementation strategies one after the other?) Or were they unpaired (in that there were separate groups that each received a different supplementation strategy)? This must be clarified.
9) Critically important, there is no table or description of the various baseline characteristics between the groups. For instance, what were the baseline characteristics of the test cohort versus the validation cohort? What about the characteristics between the five supplementation cohorts? This is essential because if differences in the mother's occurred such as more comorbidities, then they would conceivably produce less protein in their milk supply. Were they overweight? Were they malnourished? Did they have diabetes? Had they had previous babies they breastfed?
10) I would delete the comment about there is otherwise an increased need for blood checks for assessment of protein intake. This occurs several times throughout the manuscript: lines 44, 203, etc. The authors also discuss that there is a need to follow BUN and other labs in patients as each individual patient may have comorbidities that affect the absorption and/or metabolism of protein (lines238-239). These appear contradictory.
11) There needs to be an expansion on what are the limitations and strengths of this study. Specifically, is this generalizable to other settings?
Overall, this study showcases the importance of individual fortification and use of a novel equation to do so. However, there are necessary corrections that must occur prior to assuring that this data is accurate and without confounders that could sway the outcomes and the generalizability of the results.
Reviewer 2 Report
This manuscript is well written and addresses an important topic on nutrition support of preterm infants.
I have just a few suggestions:
Page 4, Table 1: the title of the table including important information and the column headings are not readable due to overlapping print.
Page 5, Table 2: in the Title the 75th quartile is mislabled as the 27th.
The authors rightly acknowledge that the BME approach needs further evaluation with regard to estimation of the percent of time this approach leads to supplemenation of protein within the desired target range. Further research with regard to growth in the NICU using this approach is also indicated. One potential problem when looking at growth outcomes is that the difference in protein intakes required to produce an effect on growth appears to need to be at least 1 g/kg/day based upon some previous studies.
Reviewer 3 Report
Dear Authors,
I believe that the following comments will help strengthen the manuscript:
Introduction:
line 37: delete the capitol letter for "breast"
lines 39-40: Please, add some more information (range values?)
line 47: Add missing dot.
General remark: If the recommendations for infants are expressed in ranges (e.g. 4.0-4.5 g / kg / d weight <1000g), are the fluctuations in protein content in breast milk large enough for standard supplementation to cause excess / oversupply of protein? Please comment.
At this point I would like to get more information why the right amount of protein is so important for babies with low birth weight; What could be the consequences of excess / deficiency?
Materials and methods:
line 63: The aim of the study should be transferred to Introduction.
lines 83-87: I would like to know more about the procedure of obtaining breast milk samples. It is very important as the content of macronutrients changing during the single feeding.
Table 1 is not unreadable.
General remark: The reader would have the benefit of including a pattern of experience in additional materials (and not just a brief description and quoting of an earlier article).
Results:
The readability of this section should be improved. Tables should be referenced in the text and not appear after a colon. Some description of results would be also helpful.
Table 3 is not fully understandable, the abbreviations used should be better explained.
There is something wrong with table 4.
Discussion:
This section is concise but logically written. However, it was difficult for me to understand from the article what is breast milk-equation (BME).
Round 2
Reviewer 1 Report
Overall, the authors have answered the previously mentioned concerns.